# Exploring the Experience of Breathlessness with the Common-Sense Model of Self-Regulation (CSM)

**DOI:** 10.3390/healthcare11121686

**Published:** 2023-06-08

**Authors:** Kylie N. Johnston, Rebecca Burgess, Slavica Kochovska, Marie T. Williams

**Affiliations:** 1Allied Health and Human Performance, Innovation, IMPlementation and Clinical Translation in Health (IIMPACT), University of South Australia, Adelaide, SA 5001, Australia; 2Faculty of Science, Medicine and Health, University of Wollongong, Wollongong, NSW 2522, Australia

**Keywords:** chronic breathlessness, common sense model, self-regulation theory, qualitative research

## Abstract

Chronic breathlessness is a multidimensional, unpleasant symptom common to many health conditions. The Common-Sense Model of Self-Regulation (CSM) was developed to help understand how individuals make sense of their illness. This model has been underused in the study of breathlessness, especially in considering how information sources are integrated within an individual’s cognitive and emotional representations of breathlessness. This descriptive qualitative study explored breathlessness beliefs, expectations, and language preferences of people experiencing chronic breathlessness using the CSM. Twenty-one community-dwelling individuals living with varying levels of breathlessness-related impairment were purposively recruited. Semi-structured interviews were conducted with questions reflecting components of the CSM. Interview transcripts were synthesized using deductive and inductive content analysis. Nineteen analytical categories emerged describing a range of cognitive and emotional breathlessness representations. Representations were developed through participants’ personal experience and information from external sources including health professionals and the internet. Specific words and phrases about breathlessness with helpful or nonhelpful connotations were identified as contributors to breathlessness representations. The CSM aligns with current multidimensional models of breathlessness and provides health professionals with a robust theoretical framework for exploring breathlessness beliefs and expectations.

## 1. Introduction

Chronic breathlessness is a distressing conscious perceptual experience constructed by the central nervous system arising from complex interactions between multiple systems responsible for breathing regulation and threat recognition [1]. While almost always pathophysiological in origin, an individual’s conscious awareness and interpretation of chronic breathlessness is a combination of somatic (bodily) signals, interpreted through cognitive (past experiences, learned associations, beliefs, expectations) and psychological (mood and affect) factors [2]. While common in a range of cardiorespiratory and neuromuscular chronic conditions, key difficulties for both people living with this adverse sensation and health professionals involved in their care is the recognition that the symptom of persistent breathlessness does not always have a consistent 1:1 relationship with markers of disease severity (e.g., pulmonary function, imaging, and oxygenation) [3], and has multifactorial causal mechanisms beyond biological impairments characteristic of specific diseases. Furthermore, effective evidence-based interventions are available for palliation of this distressing symptom even when disease management is optimized (doing “something more” rather than “nothing more”).

Current breathlessness science models and social cognitive models of illness behavior, such as the Common-Sense Model of Self-Regulation (CSM), highlight the importance of individual beliefs and expectations about breathlessness to make sense of the symptom and inform coping strategies [1,4]. The CSM is a model of illness perception that has the potential to help explain how beliefs and expectations influence coping and health-seeking or avoidance behaviors [5]. The most recent extended version of this psychometrically robust model was proposed in 2022 [4]. The CSM comprises six main components: (1) situational stimuli, (2) cognitive illness representations, (3) emotional illness representations, (4) coping strategies, (5) illness and emotional outcomes, and (6) coping appraisal. Situational stimuli refer to information sources which may be endogenous (symptom and somatic responses) or external (friends, family, health professionals, and print and social media) [4]. Cognitive illness representations reflect how an individual “makes sense” of their health condition/symptoms and includes domains of (i) identity, (ii) timeline, (iii) cause, (iv) consequences, (v) perceived control, and (vi) illness coherence [5]. Emotional illness representations reflect how an individual feels about their illness/symptom and can be activated independently or concurrently with cognitive illness representations. Cognitive and emotional illness representations are unique to each individual and are continually updated in response to memories of past experiences, new information, and sensory cues. This iterative process has been described as akin to a Bayesian process where “illness representations and prototypes represent ‘priors’ in the model while appraisals of success of coping procedures are used as posteriori information to adjust and update the representation and subsequent coping” [4].

While the CSM has been used extensively as a theoretical model to study chronic illnesses, few studies are available which use this theoretical framework to explore the symptom experience of chronic breathlessness [6,7,8]. Of these studies, situational stimuli have not been prospectively investigated, although Hallas and colleagues 2012 [6] reported the influence of health professional language with terms associated with illness diagnosis on participants emotional representations. Beliefs and expectations about chronic conditions have been shown to be influenced by language in chronic obstructive pulmonary disease [9], chronic lower-back pain [10,11], and osteoarthritis [12]. Emerging evidence indicates that health-related beliefs and expectations concerning breathlessness are influenced by the health information provided by health professionals, family, friends, and the internet [9,13,14,15,16,17,18].

The study aimed to explore breathlessness beliefs, expectations, and language preferences of people living with breathlessness in daily life. Using the CSM as a theoretical framework, the key research questions were as follows:Are there specific words or phrases (situational stimuli) relating to breathlessness that individuals do and do not prefer?Which information sources (situational stimuli) shape breathlessness beliefs and expectations?Which cognitive and emotional representations are described in relation to peoples’ experience of breathlessness?

## 2. Materials and Methods

This descriptive qualitative study was approved by the University of South Australia Human Research Ethics Committee (Protocol 201770, approved 2 April 2019). This study is reported using the Consolidated Criteria for Reporting Qualitative Research (COREQ) checklist [19]. Specific details of methodological processes are reported according to COREQ items in the Appendix A.

People were eligible for inclusion if they were (1) aged over 18 years, (2) experiencing breathlessness in daily life, (3) receiving medical management for the underlying cause of breathlessness, (4) able to comprehend written and spoken English, and (5) able to give informed consent. Purposive recruitment sought to reflect diversity (maximum variation) in the impact severity of exertional breathlessness assessed by the modified Medical Research Council dyspnea scale (mMRC) [20] (Appendix A).

There is no standardized method for estimating sample size for qualitative studies [21] and the widely used term “saturation” is increasingly contested [22]. The intent of this study was to apply an existing framework (the CSM) to consider the experience of breathlessness (situational stimuli, cognitions, and emotions). The concept of data adequacy (volume, variety, and time requirements for interpretation of data) rather than “saturation” informed our prospective sampling frame and target sample size (Appendix A). In addition, we reviewed sample sizes in prior studies reporting use of the CSM as a framework for semi-structured interview analysis, which ranged from seven to 21 participants [23,24,25,26,27,28]. Data adequacy is a recognized consideration when making a priori sample size estimates [29]. A target sample size of 20 was estimated on the basis of the higher end of sample sizes in similar prior studies, the volume of questions within the interview (responses to 22 questions), the inclusion of people with varying degrees of breathlessness (mMRC grades), and a per-participant anticipated time requirement of 10–20 h (interview plus transcription and analysis).

Participants were recruited from (1) physiotherapy practices in metropolitan Adelaide, South Australia, (2) people who had completed the Community Breathlessness Intervention Service (BLIS) program pilot study, or (3) a University of South Australia health clinic. In response to information flyers available in each recruitment site, potential participants contacted the research team to express interest or seek further information about the study. All potential participants were screened in-person or via phone or email to determine their eligibility for inclusion (Appendix A). Where participants met the inclusion criteria and indicated a willingness to participate in this study, a mutually convenient time was arranged for the study interview at either their home or at the University of South Australia City East Campus. Participants were provided with honoraria (AUD$50 gift voucher) to recompense for their time.

### 2.1. Interview Schedule

To explore participants’ experience of breathlessness, a semi-structured interview schedule was developed by the research team (R.B., M.T.W., K.N.J., and S.K.) informed by components of the CSM (situational stimuli, information sources, language preferences, and cognitive and emotional illness representations) (Appendix A; details of practice interviews and processes in Appendix A).

For each interview, a consistent member of the research team (R.B.) and the participant met for a single study visit of duration up to 1 h (details of interviewer characteristics and training in Appendix A). At the study visit, written informed consent was obtained. Participants were invited to provide demographic information and complete the mMRC [20] and a modified version of the Brief Illness Perception Questionnaire (B-IPQ) [30]. In the modified B-IPQ, the word “illness” was replaced with the word “breathlessness” and the item “How much do you experience symptoms from your breathlessness?” (identity) was removed as pilot participants indicated this item was redundant (score range 0–70, with higher scores indicating perception of greater breathlessness threat). The B-IPQ includes a single free-text item (item 8) where respondents were invited to provide three causes of their breathlessness ranked by order of importance.

The semi-structured interview was conducted in a quiet, private room and video- or audio-recorded.

Interview transcripts were emailed or posted to participants for checking no later than 1 week after the interview (Appendix A). On completion of the study all participants were provided with a plain English summary of the study findings.

### 2.2. Data Management and Synthesis

Demographic information and questionnaire data were summarized for the group and reported as frequency (%) for categorical variables (e.g., sex, living arrangements, and mMRC score) and mean (standard deviation (SD)) for continuous variables (e.g., age and B-IPQ Score). Free-text responses to B-IPQ item 8 (cause) were classified as (1) “psychological attributions”, (2) “risk factors” (within or outside of my control), (3) “immunity”, or (4) “accident or chance” according to the domains of the Revised Illness Perception Questionnaire (IPQ-R) [31]. Items that did not fall within these categories were classified as (5) “activities”, (6) “condition/anatomy”, or (7) “altered function”.

Transcripts were deidentified, and each participant was assigned an identifier code. Each interview was transcribed verbatim (RB) and checked for transcription errors by repeated reading of each entire transcript while listening to the recording. Independent checking of transcription accuracy by a second member of the research team was not undertaken. Comments were added to each transcript regarding nonverbal communication as appropriate (e.g., coughs, laughter, and the participant showing a prop or making a hand gesture).

For the CSM component of situational stimuli, each meaning unit reflecting specific words and phrases related to breathlessness was categorized according to terminology (e.g., emphysema) or descriptions (e.g., “you look well”), with subsequent codes created to reflect preference for that term (preferred or not preferred) and reason for preference under two broad categories (helpful or unhelpful). Each meaning unit reflecting a source of information about breathlessness was categorized according to the nature of the information recalled (e.g., “… told me I have COPD”), information source (e.g., respiratory specialist), and the type of information recalled (e.g., diagnosis/cause).

Content analysis was conducted according to the methodology outlined by Elo and Kangäs (2008) [32], documented within a spreadsheet (Microsoft Excel, Microsoft Corporation 2010 version 14.7.3) (Appendix A). “Meaning units” were defined as specific words, sentences or paragraphs that related to one another due to their content or context irrespective of the interview question or where the word/phrase appeared in the transcript [33].

Data in each transcript were analyzed using the following steps:(1)Open coding of meaning units using components of the CSM as an organizing framework (deductive process). During open coding, meaning units were gathered according to the relevant CSM component (i.e., situational stimuli, and cognitive and emotional representations [5].(2)Labeling of meaning units and creation of categories within CSM components (inductive process). Using a line-by-line approach, meaning units were labeled (line-by-line categories), and those that shared similar ideas/concepts within each CSM component were grouped together in descriptive categories.(3)Abstraction (inductive process). Similar descriptive categories were grouped into higher-order analytical categories.

Coding and content analysis processes were completed by a consistent member of the research team (R.B.) within 1 week of each interview. Every third interview transcript was independently coded by a second member (K.N.J.). Simultaneous data collection (R.B.) and analysis (R.B. and K.N.J.) continued during this time, at a rate of 1–2 interviews per week. Coding decisions were reviewed iteratively through meetings between the two team members throughout the data collection process. On completion of data collection and initial coding and content analysis, the research team reviewed all coding, as well as descriptive and analytical categorization, for discrepancies or areas of ambiguity. Overlapping categories and meaning units were discussed with the consensus that overlapping categories could be combined and meaning units could belong to more than one category. Consideration was given to plain language category labels and condensing data to facilitate transparent and intuitive reporting.

## 3. Results

Data collection occurred between May and September 2019. Twenty-two participants were screened for this study with 21 participants included in the final analysis (one participant declined due to family illness). Participant characteristics are presented in Table 1. This group included seven men and 14 (66.6%) women (mean age 70 standard deviation (SD) 11) and included at least one participant for each level of exertional breathlessness severity of the mMRC (range 0–4). Scores for the B-IPQ reflected a range of experiences for the severity of threat posed by breathlessness (total score mean 38.6, SD 9.6, range 13–56). There were relatively equal proportions across participants for educational attainment (primary school or lower 24%; high school 24%, certificate or diploma 28%, bachelor’s degree or higher 24%). Most participants were living with at least one other person (dual or greater occupancy 66%) and were retired from paid employment (81%).

### 3.1. Individual Preferences for Specific Words or Phrases Relating to Breathlessness

In response to a direct question, 15 participants indicated they were unable to immediately recall any specific words/phrases related to breathlessness for which they had a strong preference (for or against). Of the six participants that volunteered specific words and phrases, negative/not preferred terms were more common than positive/preferred terms. Two participants volunteered diagnostic disease labels. Table 2 summarizes participants’ preferences and explanations for/against words or phrases. Nonpreferred words/phrases were unhelpful because they evoked unpleasant emotions (e.g., fear and hate), made incorrect assumptions, did not accurately describe the experience of breathlessness, or gave nonhelpful advice. Preferred words/phrases were helpful because they accurately described the breathlessness experience or provided hope.

### 3.2. Information Sources (Situational Stimuli) Shaping Breathlessness Beliefs and Expectations

Sources of external information about breathlessness volunteered by participants included health professionals involved in their direct care and the internet. Table 3 presents a summary of descriptive categories for recalled information about breathlessness according to their source and information type. Information was more commonly recalled from face-to-face interactions with health professionals than when accessed or read on the internet.

### 3.3. Cognitive and Emotional Representations Described in Relation to Peoples’ Experience of Breathlessness

An overview of analytical categories for all cognitive and emotional representations reflecting CSM components are summarized in Table 4. Detailed tables showing all generations of analysis with unedited meaning units, line-by-line categories, descriptive categories, and analytical categories are included for cognitive representations of identity, control, and coherence in Appendix A. Details concerning evolution of analytical categories and data adequacy are presented in the Appendix A.


**Identity**


Descriptions of the breathlessness experience varied greatly and the words chosen were unique to each individual: “… oh I can’t really describe it other than that… I guess if you put… a gauze over your mouth… like be[ing] strangled… like drowning almost” [P: 2050]; “… they’re [lungs] supposed to be like a balloon… they’ve gone flat” [P: 3010]. The symptom of breathlessness was described as being misunderstood (“you almost get the feel that they’re saying ‘oh you’re just overweight and unfit’” [P: 2090]), invisible (“… people see me and I… suspect they’re thinking ‘what’s the matter with him? What a lot of bullshit, he’s fine?’” [P: 2050]), or unexplainable and isolating (“…you’re breathless… it’s not ah something that’s very explainable… with this [breathlessness] I certainly feel alone” [P: 2090]), suggesting that people who have not directly experienced breathlessness would find it difficult to imagine or relate to. Breathlessness was not the most prominent part of most participants’ identities, with many identifying themselves by their family or previous occupational roles (Appendix A).


**Consequences**


Breathlessness in daily life presented challenges for individuals. Previously enjoyed activities such as going on holidays and playing with grandchildren and everyday activities (e.g., showering and gardening) were limited (“… for the last year I haven’t played 18 holes of golf, I’ve only played nine… after nine holes walking, I just puff and puff and puff and puff…” [P: 2010]) or impossible due to breathlessness. Breathlessness limited the ability to socialize because it was exhausting (“It [breathlessness] stops me doing things I want to do. I used to like to walk to the shops and on me way talk to my neighbors and people I know and that was a big thing for me but now it’s… hard” [P: 2060]). Severe and potentially life-threatening consequences were also volunteered by participants (e.g., hospital admissions for respiratory infections and spontaneous pneumothorax). Participants reported difficulty adjusting to the fact that they could no longer do the activities that they used to and were unable to fulfil roles they previously filled with ease because of their breathlessness, which made them feel like a burden to those around them: “And I’d have to be careful who I went with [on holiday] because I’d be a bit of a burden…” [P: 2040].


**Cause**


Two components of the study session provided opportunities for participants to express their beliefs about important causes of their breathlessness: open-text responses in B-IPQ questionnaire (item 8) and specific interview questions (Appendix A). In response to the questionnaire, participants volunteered causes related to risk factors in their control (smoking: highest at 38% of respondents) or outside of their control (genetics, aging, occupation, pollution: 33%), activities involving physical exertion (29%), their medical condition or an anatomical structure (24%), altered body function (24%), or immunity (21%, Table 5). Perceived causes of breathlessness coded from the interview data overlapped in general, but were not an exact match with questionnaire data, with people attributing breathlessness to “what I do/did”, “problems with my body”, and “where I live”. Relatively few participants reported their thoughts were a cause of breathlessness, indicated in interview data only.

Participants described both causes of transient breathlessness (e.g., physical exertion) and causes of the chronic health conditions (e.g., smoking) which, in some cases, were inconsistent with a health professionals perspective (“But I know the smoking like the doctor says has … not done me any good but it wasn’t the real cause; the real cause was I had a weak chest” [P: 2060]). Almost all participants identified a specific “problem” with their body that caused their breathlessness. This included descriptions of structural or pathophysiological problems (e.g., mucous plugging and lungs constricting) or illness conditions (e.g., infection and tracheobronchomalacia), as well as aging or being overweight. Participants also identified factors to do with where they lived, local environmental conditions, and exposure to allergens as causes of breathlessness. Some participants identified stress as a cause of breathlessness: “… I’ve got to start slowin’ down ‘cause the stress level can cause breathlessness as well…” [P: 3060].


**Timeline**


Participant responses about the timeline of breathlessness were characterized by uncertainty and unpredictability. The onset of an episode of breathlessness could be sudden and unexpected: “…the onset [of breathlessness] is just so unpredictable… you could go a week and not get an attack, or you could have three or four attacks in a week…” [P: 3040]. In other participants, initial experiences of breathlessness came on gradually or were not taken much notice of: “I thought I was just getting… unfit and I didn’t make the connection… that I was actually getting a chronic disease” [P: 2050]. Responses to questions about change in breathlessness over time were highly variable, reflecting anticipated improvements, instability, steady decline, and worsening. Some participants indicated that their breathlessness would shorten or end their life: “… don’t expect a long lifespan. So it’s not gonna improve” [P: 2090].


**Control**


Control of breathlessness was achieved by a variety of prescribed pharmacological and nonpharmacological strategies, as well as through the personal behaviors of the individual (Appendix A). Participants recruited from the respiratory physiotherapy clinic frequently described how physiotherapy-led strategies helped them to control their breathlessness: “… with [respiratory physiotherapist’s name] help I could feel my breathing getting better… with the machine she gave me to… do the exercises…” [P: 3080]. Medications and oxygen were ways to manage breathlessness. Multiple personal behaviors adopted by individuals to control breathlessness included rest, stopping, pacing, distraction, exercise, avoiding triggers, self-talk, and relaxation: “… I think sometimes [sigh] if you get really breathless… if you get a bit frightened it makes you worse and so you have to say to yourself ‘stop it, relax and breathe properly’ and that probably helps…” [P: 2010].


**Coherence**


The information participants had received about their breathlessness either made sense to them Appendix A. Participants who identified they had a good understanding of their breathlessness based their understanding on their own lived experience of the condition. Participants who had experienced breathlessness in childhood often recalled this as one of the things that informed their understanding of breathlessness now. In order to ensure the information that they were receiving from their health professional was coherent, many participants said they asked questions for clarification, or asked about information they had obtained online: “… if I don’t understand I just ask… they [doctors] explain… in my language… and I do understand” [P: 2030]. Health professionals were identified as providing information that made sense to the participant and enabled them to better understand their breathlessness (“… when they’re [specialist]… explaining it [lung disease] I already knew, it kind of all made sense to me. It was good to have a figure put on it, 40 percent, ‘cause… I thought well that… explains things… but it also made me think that ‘well I better… hang on… as hard as I can… to that 40 percent…’” [P: 2050]). Participants described at times they had received information that didn’t make sense to them, or that they did not understand the explanation given by the health professional (“… I go and see the respirologist probably every 6 months. I think the most disconcerting is when I got the COPD they said ‘oh it won’t get any worse if you look after yourself’, but that’s not true I don’t think.” [P: 2050]). In other instances, the relationship between monitoring (e.g., oximetry and auscultation findings) or test results (e.g., spirometry) and the person’s experience of breathlessness was not explained or did not match up from the participants’ perspective (“… I do at home oxygen monitoring as well and that can sort of give us a clue or my team a clue of how well my lungs are at different times. Whether or not that’s got anything to do with breathlessness I’ve got no idea.” [P: 1020]).


**Emotions**


Emotions described by participants ranged from extremely low “suicidal” (after diagnosis of pulmonary fibrosis) to extremely high “uplifted” (after a respiratory specialist visit). Breathlessness was associated with predominantly negative emotions including anxiety during an unpredictable episode of breathlessness, disappointment at not being able to participate in previously enjoyed activities, and worry at the thought of breathlessness worsening toward the end of life. These negative emotions were often associated with occasions of extreme difficulty breathing, and often had reduced over time as participants learnt to cope with their breathlessness. Emotions could be highly variable from day to day depending on symptoms: “Scared some days… frightened some days… and confused and a bit… not the word depressed I think it’s more... you want to cry… ‘why is this happening to me’” [P: 3060].

Table 6 presents the synthesis of the total scores for the scalable items of the B-IPQ and final analytical categories created from the semi-structured interviews for all domains except cause (presented in Table 5). Integration of the B-IPQ and interview data (Table 5 and Table 6) shows the diversity and complexity of the interplay of domains across cognitive and emotional symptom representation domains of the CSM, which is not expressed or summarized by a higher total score on the B-IPQ.

## 4. Discussion

This study used the CSM to explore breathlessness beliefs, expectations and language preferences of people living with breathlessness in daily life. Key findings of this study were as follows: (1) few specific words and phrases about breathlessness were identified that had helpful or non-helpful connotations; (2) external sources of information about breathlessness from healthcare professionals and the internet were verified against individual’s personal experiences; (3) all domains of cognitive and emotional symptom representations informed beliefs and expectations about breathlessness. Consequences, control, emotions, and coping strategies have been explored previously in qualitative studies about breathlessness [34,35,36]. This study adds new insights into beliefs about the causes of breathlessness, timeline expectations (especially its unpredictability), individuality of the identity of breathlessness, and information sources including health professionals and their language, being integrated with personal experience to influence coherence.

### 4.1. Words and Phrases Related to Breathlessness: Room for Improvement

Most participants (15/21, 71%) did not recall words or phrases related to breathlessness that they found helpful or unhelpful. This absence may suggest the experience of invisibility of breathlessness [37] (“invisible” also evident in our findings about identity representations, Appendix A), and may have indicated a reluctance to have clinical conversations about breathlessness by patients and health professionals [38]. Where recalled, participants in this study preferred words and phrases that reflected their experience of breathlessness and offered hope. They found it unhelpful when words and phrases applied inappropriate labels, made assumptions, or provided unsuitable advice. This aligns with findings of a Delphi study with international health professional experts in chronic breathlessness [39], which recommended that conversations specifically acknowledge the individual experience of breathlessness, discuss which effective management strategies are available, clarify any misconceptions, and not blame the person or give them false reassurance. Using language that is preferred by people who experience breathlessness in daily life also adheres to principles of person-centered care, prioritizing the wishes of the person receiving healthcare over the beliefs and expectations of health professionals about their treatment [40].

### 4.2. Language of Health Professionals Is a Situational Stimulus

Language about breathlessness acts as a situational stimulus in the CSM for the development and updating of cognitive and emotional symptom representations. Words and phrases associated with chronic breathlessness may influence an individual’s perception of breathlessness, yet there is little research in this area. In other health conditions, language used by health professionals has made a substantial impact on individual’s beliefs and expectations about symptoms and illnesses. For example, research on the language associated with pain indicates that few of the commonly used medical terms relating to back pain were understood and accepted by lay people in the way health professionals intended [41]. In people with lower-back pain, words and phrases used by health professionals had a significant and lasting impact on their patients’ attitudes and beliefs, either potentially helpful or potentially harmful. Information delivered by health professionals was often interpreted as meaning that the back needed protecting, potentially leading to hypervigilance and unnecessary worry that could persist for years [42]. Using words and phrases that are at the level of an individual’s understanding (i.e., avoiding medical jargon) to explain lower-back pain can increase understanding of pain and engagement in self-management activities [43]. Further work on helpful and unhelpful explanations relevant to chronic breathlessness is indicated.

### 4.3. External Sources of Information about Breathlessness: Health Professionals Predominated

External sources of information volunteered by participants in this study included general practitioners, respiratory specialists, respiratory physiotherapists, and the internet. Health professionals were identified as key sources of information on cognitive representations of breathlessness: informing identity, cause, and diagnosis, informing control through directives and advice, influencing timeline through information about prognosis, and impacting consequences regarding assessments and measures (Table 3). In contrast, information recalled from the internet about breathlessness was sparse and grim. Findings were similar in a survey of over 4000 German adults (40% with at least one chronic condition) [44] which identified that preferred sources of health information were health professionals (general practitioners (72.1%), specialists (39.5%), and pharmacists (31.6%)) rather than the internet (31.5%), although reasons for these preferences were not reported. Subsequent to the conduct of these studies, the coronavirus disease 2019 (COVID-19) pandemic disrupted an already unpredictable diagnostic pathway for people experiencing breathlessness [45,46], which prompted an explosion of online health information and misinformation [47], a shift to remote delivery of healthcare [48], and the emergence of persistent breathlessness after COVID-19 infection [49]. Research into the impact of virtual information sources on breathlessness representations in this rapidly changing environment is indicated.

A related issue in information sources (and cognitive and emotional representations) about breathlessness is the overlap between breathlessness “the symptom” and the underlying disease condition. Information recalled from health professionals and the internet about breathlessness in this study closely linked to disease conditions and biological changes (e.g., “asbestosis”, “scarring of the lungs”, and “lung function tests”). This is understandable as (a) healthcare encounters with doctors are oriented around a diagnosis [50,51], and (b) health information resources for people living with breathlessness are often associated with a specific diagnosis. For example, a systematic review of websites providing self-management advice for people with chronic breathlessness identified 91 webpages from 44 websites with 61% (*n* = 21) of these websites specifically targeted at people with COPD [52]. Such overlapping information sources may establish and reinforce cognitive representations (beliefs) about “cause”, “prognosis”, and “timeline” of breathlessness that are anchored in physiological, environmental, or past actions, and outside of their control (Table 3, Table 4 and Table 5). In contrast, data in the “control” domain demonstrated that people had ways of palliating and managing breathlessness that were largely nonpharmacological, individualized, and inventive (Appendix A).

### 4.4. Integration of Information Sources with Own Experience, Cognitions, and Emotions

For each participant, every cognitive and emotional representation domain was important in forming a symptom representation of breathlessness to varying degree. Almost all participants volunteered information for every component, showing that the CSM is highly applicable to the exploration of symptoms including breathlessness. This study revealed the complex and highly variable individual experience of breathlessness, that may not be adequately expressed using the B-IPQ total score alone (Table 6).

In order to “make sense” of breathlessness, information from external sources was considered and interpreted alongside existing beliefs and expectations of our participants, as demonstrated by data in descriptive categories of “information received is relevant to my experience” and “I disagree with what my information sources tell me”, in the coherence domain of the CSM (Appendix A). Beliefs and expectations play a recognized role in the perception of multidimensional symptoms [53], including chronic breathlessness [54], where individuals make sense of interoceptive information on the basis of their expectations, beliefs and past experiences. In this predictive processing model, when beliefs and expectations are strongly established based on consistent past experiences, they are more influential on breathlessness perception than new information to the contrary [3,54].

Unpleasant experiences of breathlessness may form or maintain maladaptive beliefs and expectations (e.g., breathlessness means I am getting worse or damaging my body) about the symptom [55]. According to the CSM framework, such cognitive and emotional representations influence maladaptive coping strategies (e.g., activity avoidance), leading to less desirable health and emotional outcomes [5]. This suggests that health professionals should be encouraged to identify, understand, and acknowledge personal representations of breathlessness when prescribing intervention strategies, as secondary external sources of information may be less influential in decision making than existing breathlessness beliefs and expectations.

A link between control and coping strategies in the CSM was found in a systematic review by Hagger and colleagues (2017) [5], which suggested that control beliefs have a positive direct impact on problem-focused coping strategies (*p* < 0.01). This in turn leads to an indirect effect on adaptive illness outcomes, such as improved wellbeing and functioning. Many “control” strategies for breathlessness described by participants in our study could also be categorized under the domains of coping strategies, especially problem-focused coping. For example, the “control” measure of self-talk *(*“you have to say to yourself ‘stop it, relax and breathe properly’ and that probably helps…” [P: 2010]) could also be classified as a problem-focused coping strategy. The consequences and control of breathlessness were central to participants’ representations of breathlessness in previous qualitative research [34,35,36]. While individuals may have developed set of beliefs about the causes and timelines of their breathlessness, control and coping strategies and the dynamic nature of the predictive processing model [3,54] present opportunities to positively influence breathlessness perceptions.

### 4.5. Implications for Research and Clinical Practice

The CSM aligns with current multidimensional models of breathlessness [2] and provides health professionals with a framework for asking people about their breathlessness beliefs and expectations (e.g., what they think is causing their breathlessness, which management strategies they find effective and ineffective, and which words and phrases they find helpful or not). Questionnaire instruments based on the CSM, such as the B-IPQ, can be used clinically to initiate conversations about breathlessness beliefs and expectations between individuals and their health professionals.

The high applicability of the CSM to the study of breathlessness means it could be used in future research of the breathlessness experience to explore links among model components (situational stimuli, cognitive and emotional representations, coping strategies, and outcomes [4,5]). This could be achieved by analyzing effective interventions for breathlessness elements that influence cognitive and emotional symptom representations, as well as longitudinal intervention study designs to explore mediating effects of coping responses on the relationships between representations of breathlessness and health outcomes [4]. This study adds to the small amount of Australian qualitative literature on the experience of breathlessness in daily life as most qualitative studies conducted to study the experience of breathlessness to date have originated in the northern hemisphere [34,35,36]. Future studies could further explore breathlessness in more culturally and geographically diverse communities, both within Australia and internationally.

### 4.6. Strengths and Limitations

The study sampling strategy recruited people representing all five grades of breathlessness on the self-report mMRC, ensuring a diverse range of breathlessness experiences were explored in the interviews. Using a validated model (CSM) upon which to base the interview questions and validated tools (mMRC and B-IPQ) to assess breathlessness and symptom perceptions strengthened the study methodology. Follow-up phone calls were completed with every participant to determine if the transcript accurately represented the interview, if they concurred with the lead researcher’s preliminary findings, and if they gave participants an opportunity to express new information arising following the interview.

This study had several characteristics that limit the generalizability of the findings. All participants were Caucasian and resided in the wider metropolitan area of Adelaide, South Australia. Due to the lack of diversity in this study sample, the language preferences, as well as the cognitive and emotional representations, of the participants do not reflect those of other cultures and ethnicities. Over half the participants attended respiratory physiotherapy services, which may have led to a bias toward positive comments about respiratory physiotherapy. Two-thirds of participants were females, resulting in the experiences of women being more widely explored than those of men. Given that participants were also older, observed language preferences and/or beliefs and perceptions about breathlessness may not align with those of other age groups. The diagnosis and management of participants was based on self-report, not medical records, and was not verified.

## 5. Conclusions

This enquiry into information sources (including specific words and phrases), and cognitive and emotional representations confirmed the complexity of breathlessness experience. Information sources about breathlessness were influential, but potentially underused and underestimated as opportunities to update people’s representations of breathlessness. Making sense of their breathlessness experience was a dynamic process for participants who weighed up information from multiple sources across time periods. The relationships if representations with coping and health outcomes present in the CSM framework have direct and valuable applications to chronic breathlessness assessment and management. Positive coping strategies were identified in this study, in the ways people found to have control of breathlessness. Exploration of how the mediating effect of coping responses impacts threat representations and outcomes, as well as changes over time, may help explain effective interventions and generate new ways to better manage breathlessness.

## Figures and Tables

**Table 1 healthcare-11-01686-t001:** Summary of participant characteristics.

Participant	Age (Years)	Sex	Educational Attainment	Employment	Living Situation	Chronic Condition	mMRC Grade	B-IPQ Total Score
1010	65	F	P	R	1	BE, A	1	35
1020	38	F	B	S	2	A, SS	2	13
2010	74	F	H	R	2	COPD	0	33
2020	57	F	H	U	2	COPD	0	38
2030	74	F	P	R	2	BE, A, COPD	2	43
2040	78	M	B	R	1	COPD	3	40
2050	64	M	B	R	2	COPD	1	43
2060	80	F	P	R	1	A, COPD	3	52
2070	72	M	C	R	2	PF, LCa	2	35
2080	85	F	H	R	1	COPD	2	40
2090	74	F	B	R	2	IP	3	56
3010	78	M	P	R	1	COPD	1	39
3020	73	F	H	R	1	BE	2	47
3030	80	M	H	R	2	PD	2	38
3040	68	M	B	R	2	A	0	44
3050	62	F	C	R	2	A	1	29
3060	52	F	C	U	3	A	2	36
3070	84	F	P	R	2	COPD	1	24
3080	79	M	C	R	2	COPD	3	38
3090	73	F	C	R	2	A	1	36
4010	60	F	C	U	1	A, T	4	52

Educational attainment = primary school or lower (P), high school (H), certificate or diploma (C), bachelor’s degree or higher (B). Employment = retired (R), unemployed (UE), student (S). Living situation = sole occupant (1), dual occupant (2), >dual occupant (3). Chronic condition = asthma (A), bronchiectasis (BE), chronic obstructive pulmonary disease (COPD), interstitial pneumonitis (IP), lung cancer (LCa), Parkinson’s disease (PD), pulmonary fibrosis (PF), Sicca syndrome (SS), tracheomalacia (T). mMRC = modified Medical Research Council breathlessness scale (range 0–4). B-IPQ = Brief Illness Perception Questionnaire (score range 0–70, with higher scores reflecting greater severity of breathlessness threat).

**Table 2 healthcare-11-01686-t002:** Specific words or phrases recalled by participants including positive/negative connotations.

Meaning Unit	Word/Phrase	Preferred (P) or Not (N)	Reason for Preference
“Emphysema, that… word sort of frightens me a bit. I don’t like the word emphysema but anything else is just COPD” [P3070]	Emphysema	N	Unhelpful label (frightening)
“I hate that word, what’s it called? COPD. So I don’t use that. Put the blinkers on… that one” [P2010]	COPD	N	Unhelpful label (hated)
“… the biggest one that really distresses me is people telling me that I look well… you can’t snap back at them and say ‘I can’t breathe, thank you’, so I find that very distressing” [P2090]	You look well	N	Unhelpful assumption
“… ‘you must have been a smoker’ and I’ve never smoked… and I can’t even stand the smell of cigarette smoke” [P3020]	You must have been a smoker	N	Unhelpful assumption
I “… phrases you like or prefer to be used to do with breathlessness?” P “It’s not the end of the world” [P1020]	It’s not the end of the world	P	Helpful phrase (gives hope)
I “… phrases you like or prefer to be used to do with breathlessness?” P “It can go into remission” [P1020]	It can go into remission	P	Helpful phrase (gives hope)
“I think… shortness of breath is probably the best way of describing something cause it’s not just breathlessness, cause I can breathe, it’s just that I can’t get enough air into me… I’d rather someone say ‘I’ve got shortness of breath’. And see… breathlessness is associated quite often with activity and all this. Shortness of breath can come with anything or nothing” [P4010]	Breathlessness	N	Unhelpful label (doesn’t fit experience)
	Shortness of breath	P	Helpful phrase (fits experience)
I “… the most unhelpful thing you’ve been told about breathlessness?” P “That it’s anxiety. Words such as ‘look if you are not gonna cooperate with us, we can’t help you, you know’; ‘breathe slowly’; ‘slow your breathing down’… Try doin it when you can’t breathe” [P4010]	Breathe slowly	N	Unhelpful advice
	Look if you are not gonna cooperate with us	N	Unhelpful advice

**Table 3 healthcare-11-01686-t003:** Descriptions, sources and types of recalled information about breathlessness ^1^.

	Internet	General Practitioner	Respiratory Specialist	Respiratory Physiotherapist
Diagnosis/cause	- Lungs are turning to chalk	- Lungs are ruined - Asbestosis - Caused by smoking	- Says there’s nothing wrong with me - Lungs are constricting - Caused by smoking - Nodes on my lungs - Not a lot of room in my lungs - I’ve got this thing called… - Scarring of the lungs - Process of fibrosis - They don’t know why	NA
Directives/advice	NA	- Don’t let a cold settle, get straight on top of it - Rest and take medication - Go and see a specialist - Stay inside - Relax	- Recommends respiratory physiotherapy - Exercise is important - Failed interventions	- Keep breathing - Exercise is important - How treatment works
Prognosis	- Grim outlook - Impossible goals	- Nothing more they can do for me	- I’m going to die - You’re not going to get much better - Not much they can do	NA
Assessments/measures	NA	- Lung function test results - Doctor’s observations	- Lung function test results - Specialist’s observations	- Oxygen saturation

^1^ Information recalled by participants expressed as descriptive categories, coded from verbatim quotes. NA = no information volunteered.

**Table 4 healthcare-11-01686-t004:** Summary of final analytical categories for cognitive and emotional Common-Sense Model of Self-Regulation (CSM) domains including descriptions.

CSM Domain ^1^	Analytical Categories ^2^	Summary
Identity	Identity of breathlessness as a symptom	Unpleasantness of breathlessness described in unique and individual ways
Consequences	Loss of enjoyment in life	Activities previously enjoyed including sports, exercise, holidays, and family outings are difficult or impossible
	Impacts on everyday life	Every aspect of daily life negatively affected by breathlessness
	Severe impacts	Traumatic experiences of breathlessness that were life-threatening
Cause	What I do/did	Breathlessness viewed as being the result of their actions now or in the past
	My thoughts	Psychological stress seen as a contributor to breathlessness
	Problems with my body	Breathlessness due to physiological causes outside of their control
	Where I live	Environmental factors identified as contributors to breathlessness
Timeline	I did not expect this	The onset of breathlessness is not something you expect in life
	Progression. I think…	Uncertainty about how breathlessness has changed over time and how it will change into the future
	End of life	Acknowledgement breathlessness can end their life
Control	What I have learnt to do when I am breathless	Nonpharmacological strategies used to manage breathlessness
	What the doctor told me to take when I am breathless	Pharmacological strategies used to manage breathlessness
	What I do when I am breathless	What participants do when they are breathless in order to relieve it
Coherence	It all makes sense to me	What makes sense to participants about their breathlessness
	It does not make any sense to me	What does not make sense to participants about their breathlessness
Emotions	Feelings I do not want to feel	Many different negative emotions accompany their breathlessness
	There is hope	Trying to stay positive when faced with breathlessness
	Dynamic	Breathlessness causes ever-changing emotions

^1^ Deductive category; ^2^ final inductive categories.

**Table 5 healthcare-11-01686-t005:** “Cause” component of CSM: Analytical categories (indicated by √) and Brief Illness Perception Questionnaire (B-IPQ) responses (indicated by X) ranked by participants’ (*n* = 21) total B-IPQ score (lowest to highest).

		Cause—B-IPQ Domains						Cause—Analytical Categories			
Participant ID	B-IPQ Total Score	Risk Factors (In My Control)	Risk Factors (Out of My Control)	Immunity	Activities	Condition/Anatomy	Altered Function	What I Do/Did	My Thoughts	Problems with My Body	Where I Live
1020	13			X	X			√		√	
3070	24	X	X					√		√	
3050	29		X	X							√
2010	33	X	X					√		√	√
1010	35				X	X				√	
2070	35		X		X		X			√	
3060	36					X	X		√	√	√
3090	36				X					√	√
2020	38				X			√		√	
3030	38					X	X			√	
3080	38	X	X					√	√		√
3010	39	X						√			√
2040	40	X						√		√	√
2080	40		X					√			
2030	43	X					X	√		√	
2050	43	X						√			√
3040	44			X						√	√
3020	47		X		X			√		√	√
2060	52	X		X				√		√	
4010	52					X	X			√	
2090	56					X				√	
*n* (%)		8 (38)	7 (33)	4 (21)	6 (29)	5 (24)	5 (24)	12 (57)	2 (10)	16 (76)	10 (48)

B-IPQ—Brief Illness Perception Questionnaire (score range 0–70); ID—identification; *n*—number of participants.

**Table 6 healthcare-11-01686-t006:** Analytical categories for Common-Sense Model of Self-Regulation (CSM) domains (excluding Cause) ranked by participants’ (*n* = 21) total Brief Illness Perception Questionnaire (B-IPQ) score (lowest (least threat) to highest (greatest threat). Presence of category indicated by grey shading.

		Identity	Consequences			Timeline			Control			Coherence		Emotions		
Participant ID	B-IPQ Total Score	Identity of Breathlessness as a Symptom	Loss of Enjoyment in Life	Impacts on Everyday Life	Severe Impacts	I Didn’t Expect This	Progression. I Think…	End of Life	What I Have Learnt to Do When I’m Breathless	What the Doctor Told Me to Do When I’m Breathless	What I Do When I’m Breathless	It All Makes Sense to Me	It Doesn’t Make Any Sense to Me	Feelings I Don’t Want to Feel	There Is Hope	Dynamic
1020	13															
3070	24															
3050	29															
2010	33															
1010	35															
2070	35															
3060	36															
3090	36															
2020	38															
3030	38															
3080	38															
3010	39															
2040	40															
2080	40															
2030	43															
2050	43															
3040	44															
3020	47															
2060	52															
4010	52															
2090	56															
*n* (%)		16 (76)	15 (71)	17 (81)	7 (33)	13 (62)	20 (95)	6 (29)	12 (57)	17 (81)	18 (86)	17 (81)	11 (52)	17 (81)	5 (24)	1 (5)

B-IPQ—Brief Illness Perception Questionnaire (score range 0–70); ID—identification; *n*—number of participants.

## Data Availability

Detailed data (deidentified meaning units, coding, and categories) supporting the reported results can be accessed from R.B. “The use of the common-sense model of self-regulation to explore breathlessness beliefs, expectations, and language preferences”, Master’s by Research (Health Sciences), University of South Australia, 2020. UniSA Repository: https://hdl.handle.net/11541.2/146523, accessed on 7 June 2023.

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
