# Peer review of "Exploring the Experience of Breathlessness with the Common-Sense Model of Self-Regulation (CSM)"

_healthcare, 2023, doi:10.3390/healthcare11121686_

Round 1

Author Response

We thank the reviewer for their thoughtful commentary and suggestions. We have attached a file with detailed responses to both reviewers

Reviewer 2 Report

Thank you for your submission. The title and abstract immediately caught my attention and interest. I believe that this study has important implications for the issues faced by patients with dyspnea in accessing interventions. This study used a qualitative research approach using the CSM framework to explore dyspnea beliefs, expectations, and language preferences of patients with dyspnea in their daily lives. Some of my observations about this article are as follows, and I would like to commend the authors for using the COREQ inventory to report qualitative research.

1. Nineteen analytical categories emerged, no larger or smaller genera, suggesting themes and sub-themes to describe the results more clearly.

2. In qualitative studies we use "saturation" to determine the sample size, and it is necessary to further confirm or provide a basis for estimating in advance whether the sample size meets the criteria for qualitative studies.

3. Whether the data from the pilot study were included in the valid data for analysis needs to be described

4.In the inclusion criteria, it is mentioned that one of the indicators of mMRC needs to be met, but during the interview process, it is also mentioned that the mMRC and the B-IPQ questionnaire need to be completed, what is the difference between these two sessions?

5. Who completed the interviews and their relevant qualitative research experiences need to be described.

6. The transcription process also needs to be described in detail, e.g., who performed the transcription and who performed the check review work within 24 h of the interview.

7. This study lacks a description of the key feature of qualitative research, "simultaneous data collection and analysis".

8. Software to manage the data – It is applicable but not stated in the methods. Was any software used to manage the qualitative data (e.g. NVivo, Word, Excel)? How was the analysis carried out, e.g. using a table in word/excel, or by hand using traditional methods (coloured pens and post-it notes etc.).

9. The details of data saturation are not clear. For example, when does data saturation occur during the data collection or analysis process? When does data collection and analysis stop?

10. I am interested in "Information sources (situational stimuli) shaping breathlessness beliefs and expectations" and would like to know how patients' beliefs and expectations change in different contexts, and what causes the changes, perhaps in more detail in the discussion.

11. The relationship between theme and content needs to be further clarified.

12. This study found that participants preferred words and phrases that reflected their experience of breathlessness and provided hope, but the reasons behind this phenomenon were not explored in depth; why did participants have such a preference? The same question arises in the other discussion sections of the article.

13. More references are needed throughout the discussion to support your points.

  • Modify the descriptions as appropriate.

Author Response

(The authors gave the same response as above.)
